# A universal and independent synthetic DNA ladder for the quantitative measurement of genomic features

Andre L. M. Reis[1,2], Ira W. Deveson [1,2], Ted Wong[1], Bindu Swapna Madala[1], Chris Barker[1], James Blackburn [1,2], Esteban Marcellin[3] & Tim R. Mercer [1,2 ✉]

Standard units of measurement are required for the quantitative description of nature; however, few standard units have been established for genomics to date. Here, we have developed a synthetic DNA ladder that defines a quantitative standard unit that can measure DNA sequence abundance within a next-generation sequencing library. The ladder can be spiked into a DNA sample, and act as an internal scale that measures quantitative genetics features. Unlike previous spike-ins, the ladder is encoded within a single molecule, and can be equivalently and independently synthesized by different laboratories. We show how the ladder can measure diverse quantitative features, including human genetic variation and microbial abundance, and also estimate uncertainty due to technical variation and improve normalization between libraries. This ladder provides an independent quantitative unit that can be used with any organism, application or technology, thereby providing a common metric by which genomes can be measured.

[1] Garvan Institute of Medical Research, Sydney, New South Wales, Australia. [2] St Vincent's Clinical School, Faculty of Medicine, The University of New South Wales, Sydney, New South Wales, Australia. [3] Australian Institute for Bioengineering and Nanotechnology, The University of Queensland, Brisbane, Queensland, Australia. ✉email: t.mercer@garvan.org.au

Scientific inquiry relies on standard units of measurement to provide precise, reproducible, and quantitative descriptions of natural phenomena[1]. These measurements are often taken relative to reference standards with known, reliable, and often graduated properties[2,3].

A commonly used reference standard in molecular biology is the DNA ladder, which constitutes of a set of DNA fragments of known sizes that forms a ladder during migration in gel electrophoresis[4]. This DNA ladder provides a reference to determine the size of other unknown DNA fragments migrating in the gel. The use of a common ladder also allows the size of DNA fragments to be compared between experiments, and under different migration conditions.

Despite these advantages, no similar quantitative DNA ladder is available against which to measure abundance in next-generation sequencing. Natural genetic materials, such as reference human genomes or small phage genomes, do not provide a graduated scale[5–7]. Synthetic spike-in controls can be mixed at known concentrations to form a quantitative ladder[8,9]. However, errors during mixing and preparation can result in an inconsistent and inexact graduated scale that is unable to directly measure technical variation. Furthermore, these irregular errors result in differences between batches of spike-in controls.

To address these limitations, we have designed a quantitative DNA ladder that is encoded within a single synthetic DNA sequence. This single ladder sequence encodes multiple contiguous sub-sequences, each repeated at known copy numbers (cns). When sequenced, the read count from each sub-sequence is proportional to its cn within the ladder, and in a known and exact ratio to other sub-sequences within the ladder. Together, these sub-sequences form an exact scale across a dynamic quantitative range within an NGS library.

Given that no mixing is required to assemble the ladder, any observed deviation from the expected quantitative scale indicates the scale of technical variation resulting from experimental sources, such as sample amplification, library preparation bias, sampling and coverage depth, and sequencing error. Therefore, in addition to measure quantitative abundance, the ladder can also measure technical uncertainty or variability within the NGS library.

The synthetic DNA ladder is comprised of artificial sequences, independent from any reference, and can be universally applied in different samples or applications. Within this study, we use ladders in metagenome analysis and human genome sequencing, where they provide a quantitative framework for reference-free genome analysis. Finally, we show how the synthetic ladder achieves best-in-class library normalization, and thereby enables more accurate quantitative comparisons between samples and more efficient data sharing between laboratories.

## Results

**Design of synthetic DNA ladder.** The synthetic DNA ladder is a single continuous DNA molecule that encodes multiple unique and artificial sub-sequence elements that are repeated at known cns (1×, 2×, 4×, and 8×; Fig. 1a). When the DNA ladder is sequenced, the read count for each sub-sequence will be proportional to its repeat cn, and in known and constant ratio to the other sub-sequences within the same molecule (Fig. 1a and Supplementary Fig. 1a).

The ratio between the successive sub-sequence elements in the DNA ladder can form a graduated scale, and define a quantitative standard unit, that we term cn (Fig. 1a). In a hypothetical "perfect" library that is unaffected by experimental variables, a comparison of the read counts to cn will assemble a linear and graduated quantitative ladder structure, that can be used as an internal scale for the quantification of DNA sequences in the NGS library (Fig. 1c).

To build the synthetic DNA ladder, we first designed the individual sub-sequence elements. Each element was 600 nt in length and without homology with any known natural nucleotide sequences (>25 nt, Supplementary Fig. 1b; see "Methods"). The nucleotide composition of each element is similar to ranges found in natural organisms, even though each element is distinguishable from natural DNA (Supplementary Fig. 1c; ref. [10,11]).

The artificial sequence elements were then assembled at one, two, four, or eight copies to form a single continuous ~10 kb sequence (Fig. 1a). Additional intervening and flanking sequences were also included to mitigate edge effects. Using this approach, we manufactured 14 different synthetic ladders, which were then combined at equal abundance to form a mixture. The ladder mixture can then be added to a DNA sample prior to library preparation and sequencing (Fig. 1b). In the resulting library, sequenced reads derived from the DNA ladder can be easily partitioned and analyzed given their lack of sequence homology to natural DNA (Fig. 1b).

**Using the synthetic DNA ladder in next-generation sequencing.** To initially demonstrate the performance of the synthetic DNA ladder without confounding experimental variables, we first simulated an NGS library derived from the ladder (see "Methods"). By plotting the median k-mer count per sequence element relative to cn, we reproduced the anticipated ladder structure, with a strong linear relationship between k-mer counts and cn ($R^2 = 0.9799$) and accurate and constant median ratios between successive cn (mean = 1.991, SD = 0.1490; Fig. 1c).

We next demonstrated the performance of the synthetic DNA ladder in an experimental NGS library that is subject to technical variation. Sources of variation include library preparation biases and sequencing errors that accumulate during experimental protocol. We prepared and sequenced an NGS library from the neat mixture of synthetic DNA ladders, and then plotted the observed k-mer counts for each sequence element relative to the expected cn. Due to the additional technical variation introduced during experimental steps, the observed DNA ladder had a weaker linear relationship ($R^2 = 0.8708$) and greater variation in median cn ratios (Mean = 2.034, SD = 0.3950) than the simulated library (Fig. 1c).

We finally compared the performance of the synthetic DNA ladder to spike-in controls previously designed for use in metagenomics (Supplementary Fig. 2a, b; ref. [9]). Unlike the DNA ladder, these spike-in controls form a quantitative scale by mixing individual DNA molecules at different concentrations, which can result in mixing errors. We see that the proportional variability at different cns in the ladder is on average twofold lower compared to the spike-ins (Supplementary Fig. 2c, d). Furthermore, the DNA ladder also has more accurate ratios between subsequent cns compared to the spike-ins (Supplementary Fig. 2e). This demonstrates the DNA ladder provides a more exact quantitative scale than comparable spike-in controls, and eliminates confounding technical variables that are often introduced during mixture preparation of other comparable spike-in controls.

**Measuring technical error with the synthetic DNA ladder.** Like other spike-in controls, the synthetic DNA ladder reflects the technical variation that accumulates during experimental steps of library preparation and sequencing. This is apparent in the symmetric, unimodal distribution of k-mer counts at each level of cn, which is broader in the experimental library than the simulated library (Fig. 2a). We also observe less uncertainty at higher

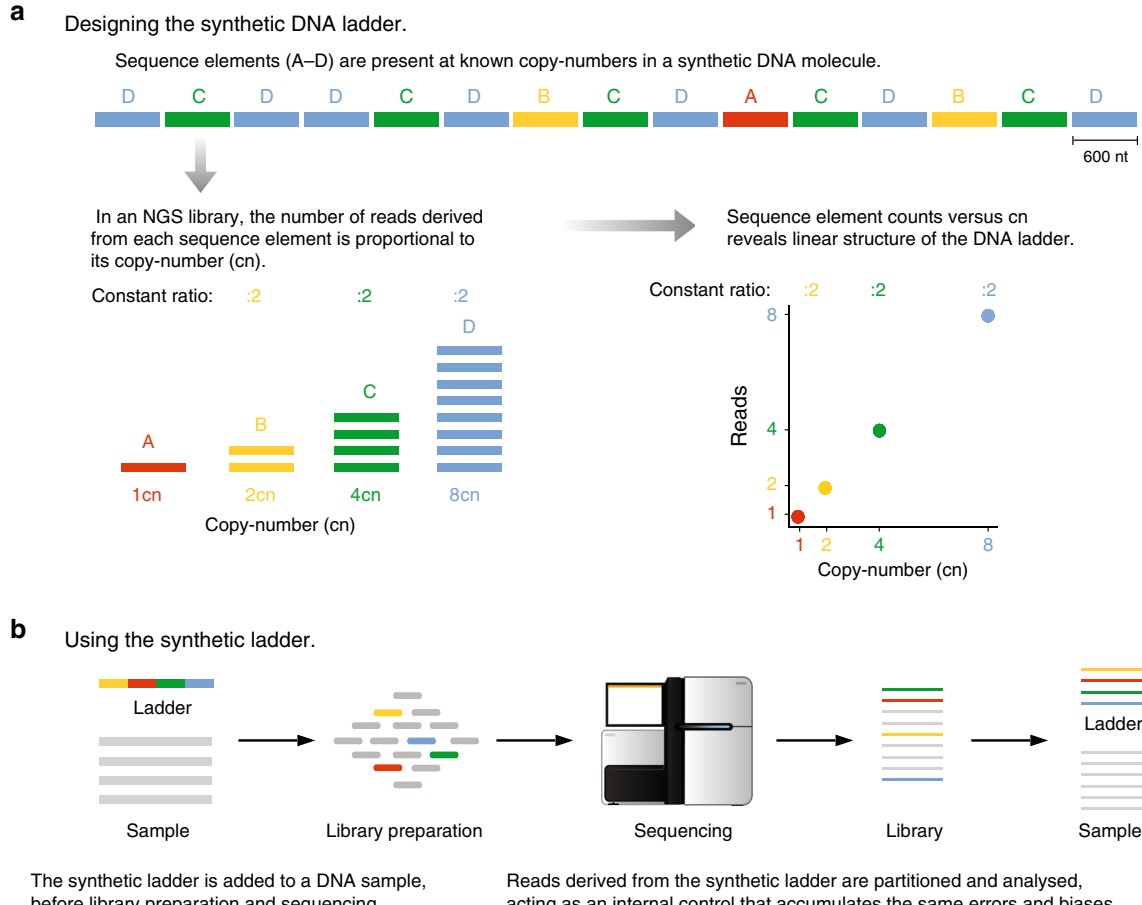

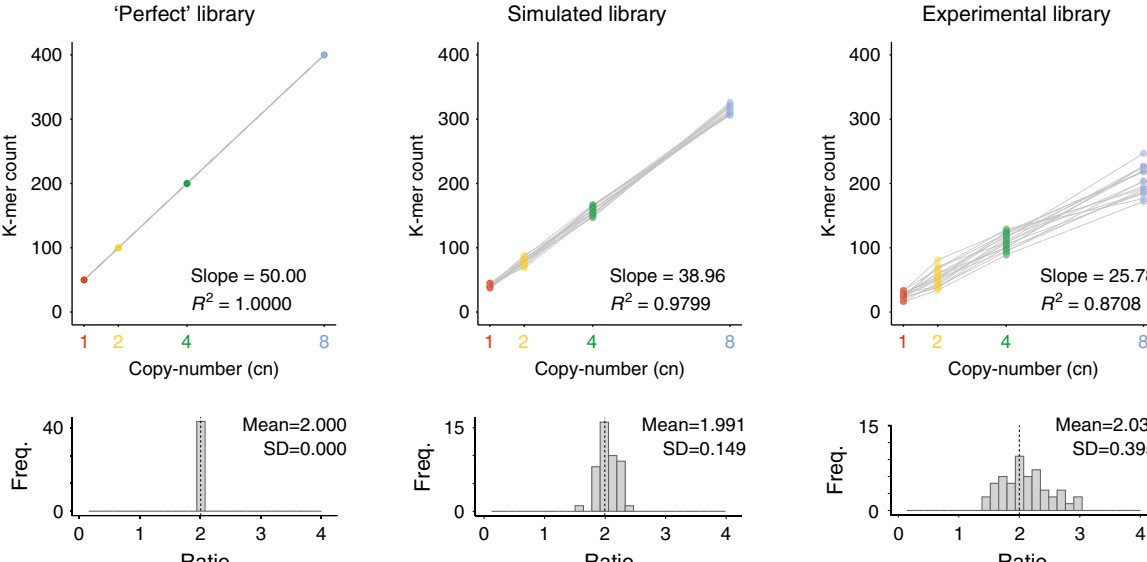

levels of cn, reflecting their greater sampling coverage and confidence (CV; 1 cn = 0.16, 2 cn = 0.12, 4 cn = 0.08, 8 cn = 0.05; Fig. 2a). The intersections between the distributions indicate the fraction of k-mers with counts that do not discriminate between successive cns (Supplementary Fig. 3). The overlap, which is also indicative of uncertainty, increases with lower depth of coverage or increased sequencing error (Supplementary Fig. 3).

We next considered whether the ladder could proportionately measure common technical errors, including sequencing error, library depth, and complexity. We first modeled the impact of sequencing errors on the ladder, finding errors caused an exponential reduction in slope, whilst also increasing variability at cn units (Fig. 2a, b and Supplementary 4a). We also evaluated library depth, finding that greater library depth proportionately

**Fig. 1 Design and use of synthetic DNA ladders. a** A synthetic ladder is a single DNA molecule that contains unique, artificial sequence elements repeated at known copy numbers (cn); A = 1 cn, B = 2 cn, C = 4 cn and D = 8 cn. The number of reads derived from each sequence element in an NGS library will be proportional to the known copy number and the abundance ratio between subsequent copy numbers is constant. **b** A mixture containing multiple synthetic DNA ladders is added to a DNA sample before library preparation and sequencing. In the output library, reads deriving from the DNA ladders can be distinguished by their artificial sequence. **c** (Top panel) Scatter plots indicate the observed abundance (in k-mer counts) of sequence elements of 14 different synthetic DNA ladders versus copy number in three different contexts: a hypothetical "perfect" library, a simulated library, and an experimental library (prepared with Nextera XT kit and sequenced on Illumina HiSeq 2500). (Bottom panel) Histograms show the distribution of ratios between subsequent copy numbers in the 14 synthetic DNA ladders that were manufactured. For the experimental library, the n = 14 independent synthetic ladders where examined over 5 independent experiments with comparable results (see Nextera XT libraries in Supplementary Figs. 5 and 6).

increased the ladder slope and decreases cn count variability (Fig. 2b–b and Supplementary Fig. 4c). We found that ladder linearity was maintained down to fivefold coverage, and provides an accurate quantitative scale across the eightfold dynamic range (Supplementary Fig. 4d). We finally observed that library complexity, as measured by unique read fragments, does not impact the ladder slope but increases count variability at different cn (see "Methods"; Fig. 2b, c and Supplementary Fig. 4b).

**Using the synthetic ladder with different sequencing technologies.** The synthetic DNA ladders provide a simple and independent metric that can evaluate sequencing technologies. To benchmark ladder performance in various experimental contexts, we prepared and sequenced ladders using different NGS applications (whole-genome sequencing or target-enriched), sequencing technologies (Oxford Nanopore MinION™, PromethION™, Illumina HiSeq 2500™, HiSeq X Ten™ and NextSeq 500™) or library preparation protocols (KAPA Hyperplus/PCR-free™, PCR-based™, and Illumina Nextera XT™; see "Methods"; Fig. 2c, Supplementary Figs. 5 and 6).

The synthetic ladder can measure the quantitative performance of the different library preparation methods, with a full and detailed evaluation provided in Supplementary Fig. 7. For example, the Illumina Nextera XT™; preparation, which uses lower input DNA amounts and additional amplification steps, exhibits a lower slope and higher variability in cn counts compared to alternative PCR-free library preparation methods that also exhibit stronger linearity (Supplementary Figs. 5 and 7)[12]. Nanopore libraries, where sequencing error is higher, also exhibit a reduced ladder slope (6.49 and 4.22 vs 35.03; Supplementary Fig. 7b) and increased count variability (Supplementary Fig. 7e)[13]. However, reducing the k-mer length mitigates the impact of sequencing error, and achieves better quantitative results using nanopore long-read data (Supplementary Fig. 8).

Target enrichment provides greater sequencing coverage at genomic regions of interest, however, new sources of variation are introduced with the additional hybridization and amplifications steps[14]. To measure this variation, we generated manufactured gene panels containing custom probes capable of capturing the ladder (see "Methods"). Notably, target-enriched libraries exhibit greater quantitative accuracy, as indicated by the greater resolution between high levels of cn (8/4 cn ratio; mean = 6.09 and SD = 2.63) compared to WGS PCR-free libraries (8/4 cn ratio; mean = 1.99 and SD = 0.25; Supplementary Fig. 6). However, this greater quantitative accuracy is confounded by a nonlinear bias resulting from target enrichment and PCR amplification bias apparent in increasing ratios between successive cn units (Supplementary Figs. 5 and 6).

**Reference-free human genome analysis with the DNA ladder.** Reference-free genome analysis, wherein individual reads or k-mers are used to make quantitative and qualitative inferences about DNA samples, avoids errors due to misplaced, or

ambiguous alignments to a reference assembly, and is required when no reference assembly is available[15–18]. However, it can be challenging to reliably measure quantitative changes within and between samples in the absence of a reference genome assembly[19]. Therefore, we next considered whether the ladders can provide a quantitative framework for measuring differences between samples during reference-free genome analysis.

To demonstrate the use of the ladder in genome sequencing, we first spiked the ladder into a well-characterized reference human DNA sample (NA12878) and performed whole-genome sequencing (see "Methods"). We then performed k-mer counting on the human genome sample and ladder (see "Methods"). The ladder was then calibrated to the accompanying NA12878 genome, so the 2 cn unit matched the median count of the diploid human genome (Fig. 3a). After calibration, 90.1% of k-mers derived from the NA12878 genome were encompassed within the eightfold dynamic range of the ladder.

We next used the known homo- and heterozygous variants annotated in the NA12878 genome to evaluate the accuracy of quantitative measurements with the ladder[6]. We found that the quantification of 1 cn and 2 cn graduations in the ladder (mean = 16.9 and 32.9) closely matched the quantification of annotated homo- and heterozygous variants (mean = 16.2 and 31.4; Fig. 3b). Furthermore, the variation in homo- and heterozygous variant counts (SD = 6.5 and 9.9) also matched the observed variation in 1 cn and 2 cn counts (SD = 5.1 and 9.4; Fig. 3b). This shows how the ladder can accurately measure quantitative genetic features in the accompanying human genome samples, which was reproducible across four technical replicates (Fig. 3c).

To demonstrate whether the ladder can be used to identify significant biological differences between genomes and in the absence of technical replicates, we next performed whole-genome sequencing of the Ashkenazi Jewish family trio; NA24149 (father), NA24143 (mother), and NA24385 (son)[20]. The resulting libraries were sequenced at comparable depth, and the ladder calibrated to the human diploid genome (Supplementary Fig. 9). These human genome samples share a large number of germline variants present at known diploid and haploid abundances, which have been extensively characterized with high confidence. Therefore, these genomes comprise a large, reliable dataset with both control and test k-mers at same and differing quantities, which can be used to validate the quantitative accuracy and fidelity of 1 cn and 2 cn within the ladder.

We then detected known fold differences in the frequencies of consanguineous variants between the family trio of genomes. This detection of fold differences was according to; (1) raw fold changes or (2) performing a pairwise one-sided t-test to detect fold differences using technical variation (standard deviation) estimated from the ladder (Fig. 4a; see "Methods"). Between the mother and son genomes, we found that using the ladder to inform a pairwise one-sided t-test, we detected differences in germline variants with a sensitivity of 0.81 and specificity of 0.79 (FDR corrected p value ≤ 0.05; Fig. 4b). Using the ladder to inform a pairwise t-test was a better discriminator of

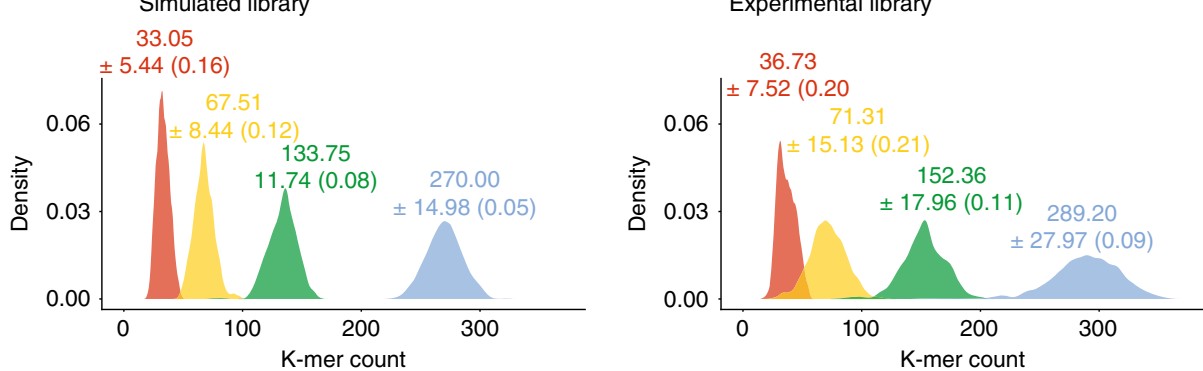

**a** Measuring technical variability at the cn units of the DNA ladder.

Simulated library

Experimental library

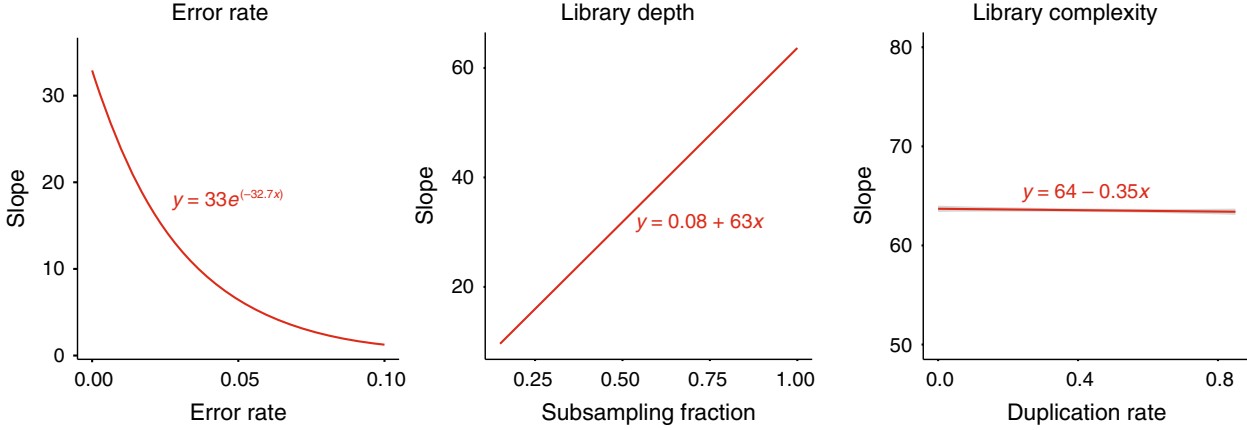

**b** Impact of different technical variables on the DNA ladder.

Error rate

$y = 33e^{(-32.7x)}$

Library depth

$y = 0.08 + 63x$

Library complexity

$y = 64 - 0.35x$

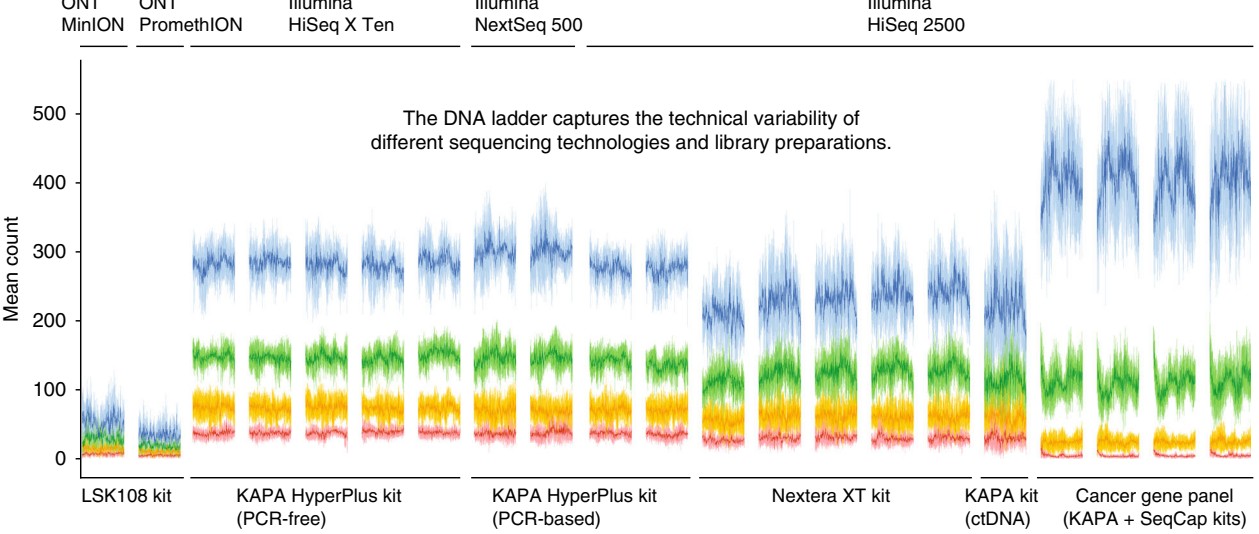

**c** Benchmarking sequencing technologies.

The DNA ladder captures the technical variability of
different sequencing technologies and library preparations.

fold-change differences (AUC = 0.84) than raw fold changes alone (AUC = 0.63; Fig. 4c). The similar results were also found for the other pairwise comparisons between father–son and father–mother genomes (Supplementary Figs. 10 and 11), which shows how in the absence of technical replicates, the detection of quantitative fold differences between genomes can be improved using estimates of technical uncertainty derived from the ladder.

**Using the ladder to normalize between metagenome samples.** Normalization can minimize technical differences and enable more accurate detection of biological differences between libraries. However, different normalization methods rely on different assumptions, with most assuming that the samples being compared have a similar composition with only a few minor differences[21]. When these assumptions are violated, such as when global bias or major quantitative differences exist, normalization

**Fig. 2 Measuring technical variation using DNA ladders. a** Density distribution of k-mer counts for each cn unit ($n = 570$ k-mers per unit; 1 cn = red, 2 cn = yellow, 4 cn = green, and 8 cn = blue) in a simulated and experimental libraries (prepared with KAPA HyperPlus PCR-free and sequenced on Illumina HiSeq X Ten), with the mean, standard deviation and coefficient of variation indicated. For the experimental library, the $n = 14$ independent synthetic ladders where examined over 5 independent experiments with comparable results (see HiSeq X Ten/PCR-free libraries in Supplementary Figs. 5, 6, and 7a). **b** Modeling the impact of common NGS technical variables such as error rate, library depth, and library complexity on the regression slope of the synthetic DNA ladder. **c** The panel illustrates the variability associated with cn units across a range of libraries generated with different sequencing technologies (e.g., ONT Nanopore MinION, ONT Nanopore PromethION, Illumina HiSeq X Ten, Illumina NextSeq 500, and Illumina HiSeq 2500 instruments) or prepared with alternative protocols (e.g., KAPA, KAPA HyperPlus PCR-free/PCR-based and Nextera XT kits, or target enrichment by oligonucleotide hybridization). The $y$ axis indicates k-mer counts, whilst the $x$ axis indicates the position across the sub-sequence length (600 nt). The mean k-mer count (opaque line) and standard deviation (transparent line) are determined across all $n = 14$ independent synthetic ladders in all the sequencing experiments.

can be inaccurate[22]. In contrast to natural samples, which are often unpredictable, the ladder retains an invariant composition in each sample, and thereby complies with the underlying assumptions of multiple normalization methods.

To investigate the advantages of using the ladder for normalization, we first generated multiple mock microbial communities by mixing together extracted genomes at known concentrations (community A, B, and C; see "Methods"). These provide a useful set of reference mock communities, with known fold differences in microbial genome abundance. Differences in genome abundance between communities was either balanced (where the sum of log fold changes is zero), or unbalanced (where the sum of log fold changes is nonzero; Fig. 5a and Supplementary Fig. 12a). Synthetic DNA ladders were added to the DNA mixture of each community, before library preparation and sequencing (see "Methods"). In addition, we also generated matched simulated libraries to aide analysis (see "Methods").

We first performed normalization based solely on the microbial samples (i.e., without reference to the ladder) using a range of common normalization methods, including median of ratios (MR[23]), trimmed mean of $m$ values (TMM[24]), and upper quartile normalization (UQ[25] Fig. 5b, Supplementary Figs. 12a and 13b). We then assessed normalization performance according to the detection of known fold differences between the mock communities. In both simulated and experimental libraries, normalization between unbalanced communities was poor and inconsistent, and artefactually underestimated fold-change differences (Fig. 5c and Supplementary Fig. 12b).

To address these errors, we next used the ladder to perform normalization between samples (using MR, TMM, and UQ methods). In these cases, normalization scaling factors were calculated based on the ladder, and then applied to the accompanying microbial sample (Fig. 5b, Supplementary Figs. 12a and 13b). Using this approach, we found that all normalization methods (MR, TMM, and UQ) achieved superior detection of known fold differences between communities when trained using the ladder for both simulated libraries (with ladder AUC = 0.91, without ladder AUC = 0.80) and experimental libraries (with ladder AUC = 0.86, without ladder AUC = 0.70; Fig. 5c and Supplementary Fig. 12b).

The use of the ladder also avoids technical artefacts that arise from normalization of differing library structures. For example, we found the ladder does not change the expected fold-difference relationships between unbalanced communities A and B, while normalizing samples directly (with any of the methods) underestimates the abundance fold differences (Fig. 5d and Supplementary Fig. 12c). Together, this indicates that the use of the ladder can achieve best-in-class normalization performance, enabling more accurate comparison between different samples and NGS experiments.

**Discussion**. Within this study, we developed synthetic DNA ladders that can define a quantitative measurement unit (that we

term the cn) within an NGS library. We demonstrate the use of the synthetic DNA ladders in various applications, including evaluating different sequencing technologies, measuring genetic variation between human DNA samples and comparing the composition of different microbial communities. In each case, the DNA ladders reliably measured abundance of DNA sequences within NGS libraries, and improved quantitative comparisons.

We show that quantitative measurements using the ladder are commutable to the accompanying DNA sample. Consequently, the detection of fold differences within and between libraries can be markedly improved by incorporating estimates of variation derived from the ladder. Furthermore, technical variation can be estimated even in the absence of technical replicates, which is typical of genome sequencing experiments. Synthetic DNA ladders can also be used with any organism or sequencing technology and provide a simple and sensitive metric for evaluating library quality and benchmarking different reagents, equipment, and methods.

The DNA ladders provide several benefits over previous spike-in controls, that are typically assembled by mixing different synthetic DNA molecules at known concentrations, and variation in this mixing process causes batch-specific differences between different spike-in controls mixtures[9,26]. By contrast, the quantitative units defined by the DNA ladder are encoded within a single sequence, thereby ensuring accurate stoichiometry and do not require mixing. The sequence alone is sufficient for an independent laboratory to synthesize a functionally equivalent DNA ladder. In addition, given the diverse range of k-mers present in the synthetic ladders, they may be used as an in-run reference against which to measure sequencing errors, improve base-calling algorithms, and normalize sequencing biases.

DNA ladder sequences can be easily standardized, stored, and transmitted to different laboratories for decentralized and sustainable manufacture. These ladders that are synthesized by different laboratories will be functionally equivalent, with standardized and interoperable measurement units. These ladders can then enable the accurate normalization of genomic data between the different samples, thereby aiding efficient data sharing and comparisons between laboratories. As genome data accumulate at an increasing and unprecedented scale, and sequencing technologies diversify, the ladder provides a standardized quantitative metric with the potential to act as a common denominator and ensure interoperability across genomics.

## Methods
**Design and production of synthetic ladders.** Each ladder has four different sub-sequence elements at different cns. We set the length of sub-sequence elements to 600 nt so they were longer than the typical read or fragment length observed in short-read library preparation protocols. To obtain the sub-sequences, we selected 600 nt long sequences from hg38 which were the shuffled to remove homology. To confirm the artificiality of our sequences, we performed a blast search against the nonredundant nucleotide database (nr/nt)[27], which found no exact matches longer than 25 nt (Supplementary Fig. 1). We randomly organized 4 sub-sequences into

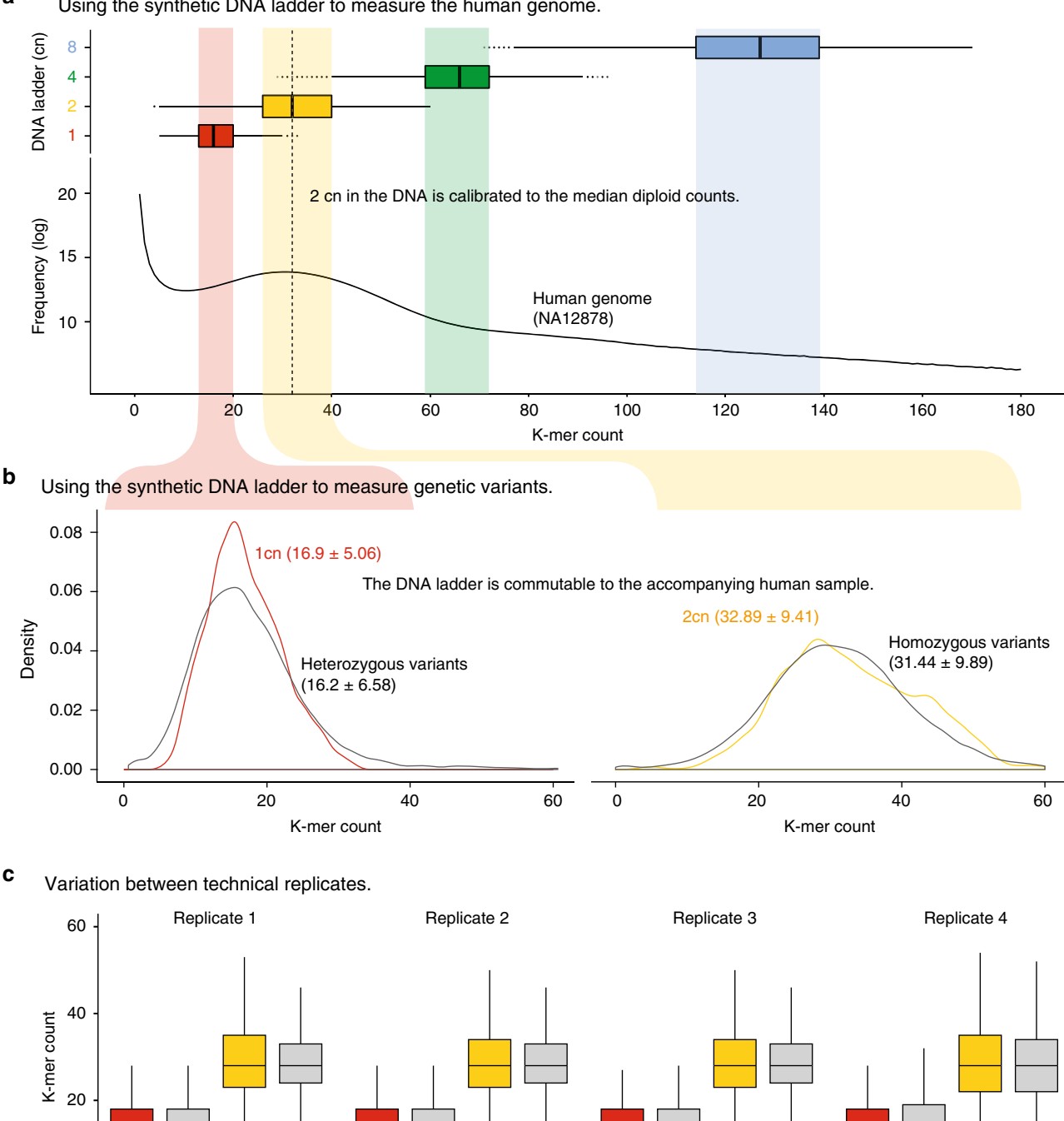

**Fig. 3 Using the DNA ladder to measure quantitative features of the human genome. a** Box-whisker plots indicate the k-mer count distribution associated with each cn unit (upper panel) in comparison to the k-mer count distribution for the accompanying human NA12878 genome (bottom panel). **b** Density plot shows the k-mer count distribution for 1 and 2 cn units that exhibit a commutable distribution to heterozygous and homozygous k-mers in the NA12878 genome. **c** Box-whisker plots show the variation in k-mer count distribution in the DNA ladder and NA12878 germline variants across four replicates. The median absolute deviation for each replicate is indicated below. The results observed in (**a**) and (**b**) were reproduced over four independent experiments as can be observed in the boxplots of (**c**). In (**a**) and (**c**), where data are represented as boxplots, the middle line is the median, the lower, and upper hinges correspond to the first and third quartiles, the upper whisker extends from the hinge to the largest value no further than 1.5 × IQR from the hinge (where IQR is the inter-quartile rage) and the lower whisker extends from the hinge to the smallest value at most 1.5 × IQR of the hinge and any points beyond the whiskers are represented individually.

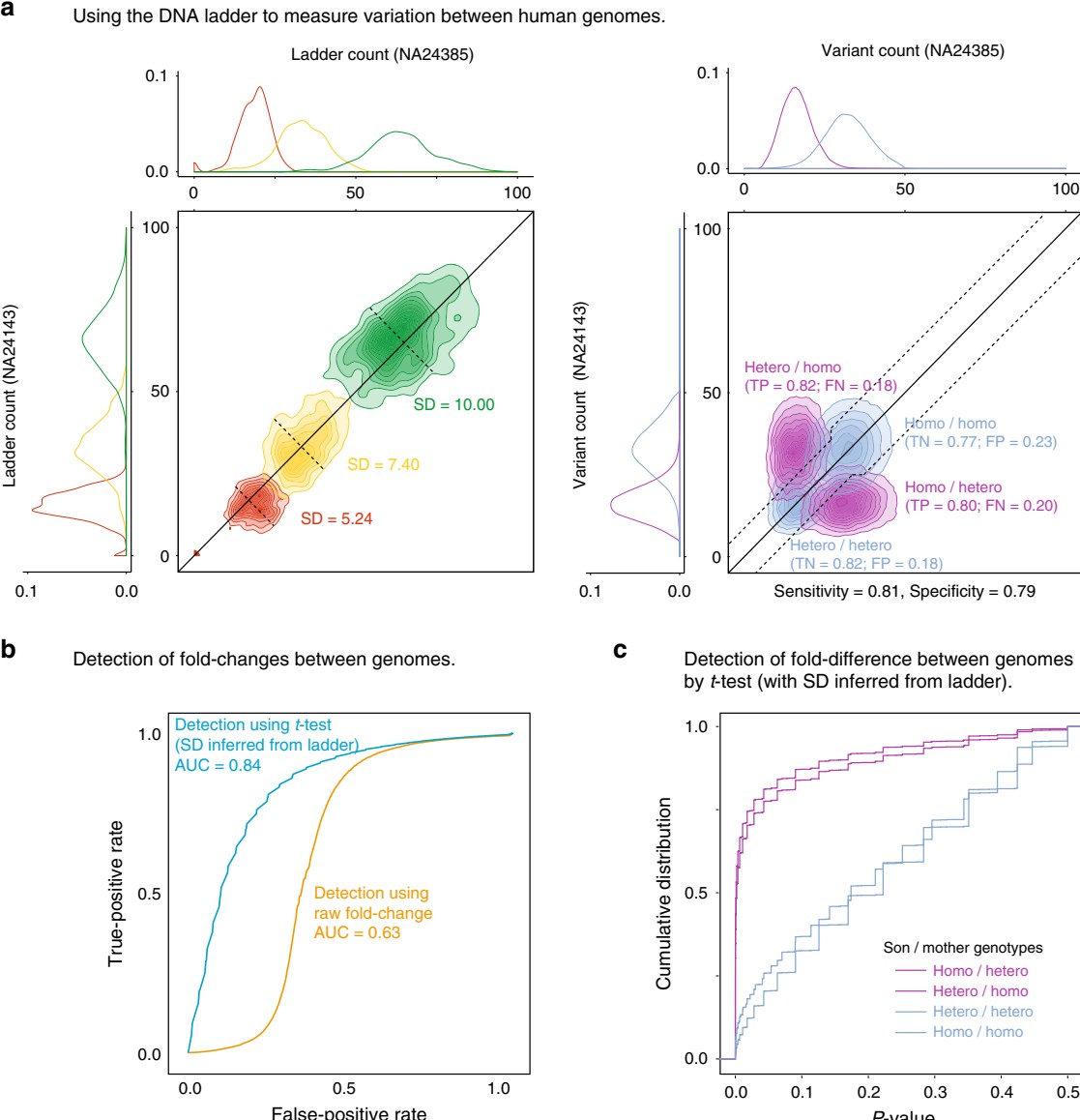

**Fig. 4 Using the DNA ladder to detect variant differences between family trio of human genomes. a** Scatter plots compare the (left) ladder k-mer counts and (right) variant k-mer counts in the son (NA2385) and mother (NA24143) genomes. The dashed line (right) indicates the significance threshold ($p = 0.05$; with SD inferred from ladder) for discriminating variant differences between the two genomes. **b** ROC curves ranking the detection of variant differences by raw fold-change (orange) or t-test significance (light blue; one-sided t-test uses SD estimated from the DNA ladder). **c** Cumulative distributions of known fold differences between genome variants, as ranked by p value significance (derived from ladder), shows effective discrimination between true- from false-positive differences. In (**a**)–(**c**), the statistical test used was a one-sided t-test and p values were adjusted for a false discovery rate (FDR) of 0.05. Independent comparisons between son/father and father/mother also achieved comparable results (see Supplementary Figs. 10 and 11).

14 ladders by joining the sub-sequences according to the cns 1, 2, 4, and 8. An intervening unique artificial sequence of 20 nt was added between each sub-sequence element and a 500 nt common sequence was added to 5′ and 3′ end of each ladder to mitigate potential edge effects.

The ladder sequences were synthesized by a commercial vendor (ThermoFisher-GeneArt) and cloned into vectors. The plasmids containing the ladders were transformed in *Escherichia coli*, then grown in a 50 ml culture and later purified. Each ladder excised from the plasmids; then the size of the final ladder sequence was confirmed on an agarose gel and quantified by UV fluorometry (Thermofisher Qbuit). In total 14 different DNA ladders were generated.

Purified DNA ladders were quantified using the BR dsDNA Qubit Assay on a Qubit 2.0 Fluorometer (Life Technologies) and verified on the Agilent 2100 Bioanalyzer with the Agilent High Sensitivity DNA Kit (Agilent Technologies). Individual DNA ladders were combined at equimolar concentrations using an epMotion 5070 liquid handling robot. Mixture stocks were prepared as single-use aliquots and stored at −80 °C.

**Simulation of sequencing libraries**. We simulated an Illumina paired-end library of a neat DNA ladder mixture using the Wgsim software (version 1.9[28]). The DNA ladders sequences were used as templates for the simulation and each DNA ladder contributed the same number of reads in final library to ensure equimolar concentration. The parameters used for error rate, insert length and insert length standard deviation were based on an experimental library.

**Preparation of DNA libraries and sequencing**. We first sequenced five neat preparations of the DNA ladder mixture. Libraries were prepared using the Nextera XT Sample Prep Kit (Illumina) in accordance to the manufacturer's instructions. Prepared libraries were quantified on Qubit (Invitrogen) and verified on Agilent 2100 Bioanalyzer with the Agilent High Sensitivity DNA Kit (Agilent Technologies). Finally, libraries were sequenced on a HiSeq 2500 (Illumina) producing paired reads of 125 nt at the Kinghorn Centre for Clinical Genomics, Darlinghurst, NSW. We prepared two additional replicates of DNA ladder neat mixtures with KAPA HyperPlus PCR-based kit. After quantification and quality assessment with

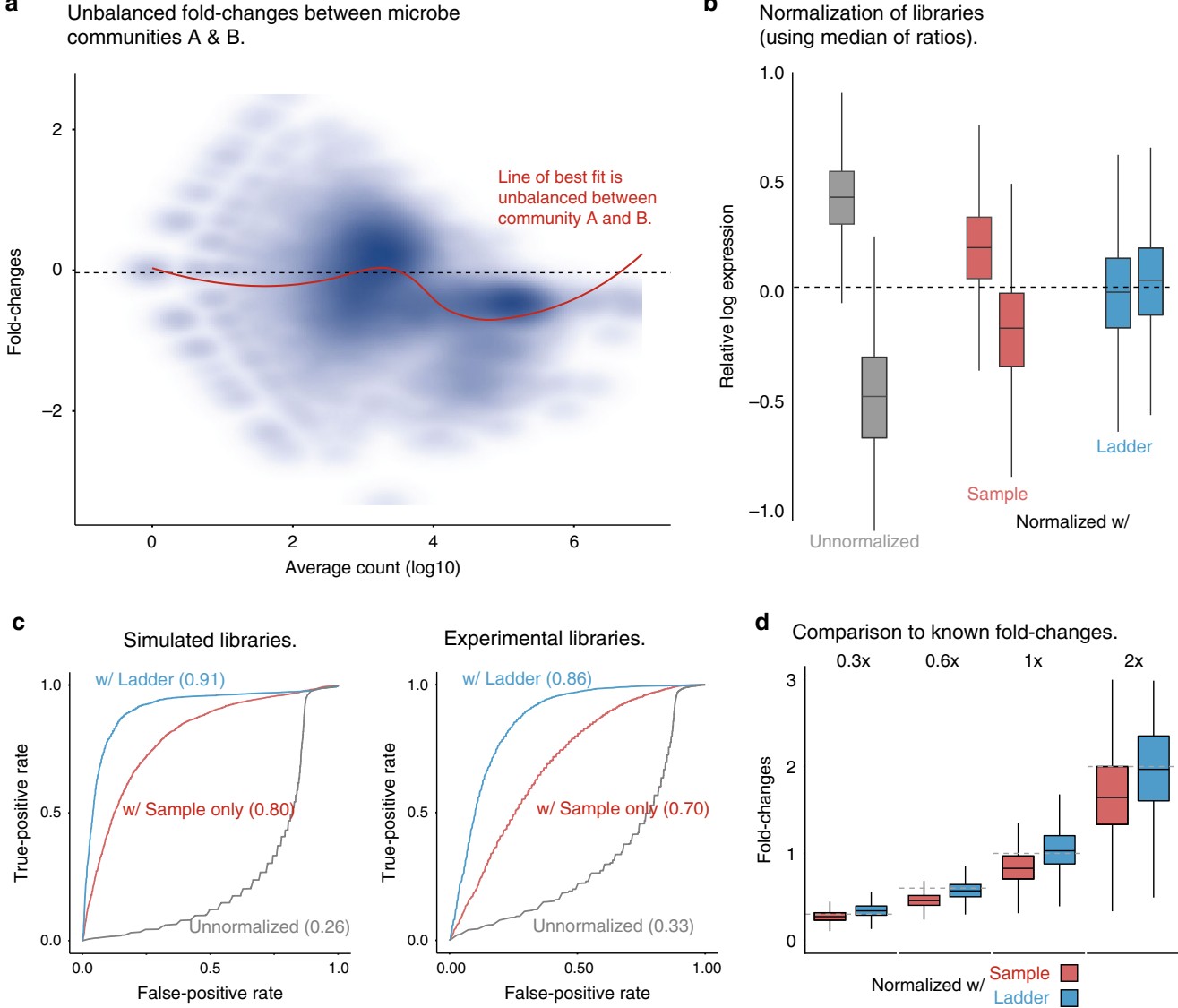

**Fig. 5 Using the DNA ladder as a reference to normalize between metagenomic samples. a** Scatter-plot of log fold changes between communities A and B, with the line of best fit (red) indicating unbalanced differences between them. **b** RLE plots illustrate the impact of normalizing the libraries based on the sample (red) or the DNA ladder (blue) using median of ratios method. **c** ROC plots indicate the detection of known fold-change differences between the mock communities following normalization with synthetic ladder (blue), sample (red), or unnormalized (gray) in simulated or experimental libraries. **d** The boxplots show the observed k-mer count fold differences between communities A & B following normalization with the DNA ladder (blue) or sample (red) using median of ratios method, at different expected fold changes (0.3, 0.6, 1, and 2) represented by horizontal gray lines. A total of 19,768 k-mers from 9 different bacterial species (see "Methods") mixed at known fold changes were used to evaluated the normalization performance between communities A` & B. In (**b**) and (**d**), where data are represented as boxplots, the middle line is the median, the lower, and upper hinges correspond to the first and third quartiles, the upper whisker extends from the hinge to the largest value no further than 1.5 × IQR from the hinge (where IQR is the inter-quartile rage) and the lower whisker extends from the hinge to the smallest value at most 1.5 × IQR of the hinge and any points beyond the whiskers are represented individually.

Qubit (Invitrogen) and Agilent 2100 Bioanalyzer with the Agilent High Sensitivity DNA Kit (Agilent Technologies), the libraries were sequenced on NextSeq 500 (Illumina) producing paired reads of 150 nt at the Kinghorn Centre for Clinical Genomics, Darlinghurst, NSW.

We next prepared libraries by adding the DNA ladder mixture to human genomic DNA samples. Immortalized lymphoblast cell lines derived from GM12878 and the Ashkenazi Jewish trio (GM24385, GM24149, and GM24143) human individuals were purchased from Coriell Biorespotory (https://catalog.coriell.org/). The cell lines were cultured independently according to Coriell Cell Repositories growth protocols and standards. DNA was extracted using TRIzol (Invitrogen) according to the manufacturer's instructions. Extracted DNA samples were treated with RNase A, followed by cleanup with Genomic DNA Clean & Concentrator kit (Zymo Research). Purified DNA was quantified using Nanodrop (Thermo Scientific). The DNA ladder mixture was added at 1% relative concentration in each of the genomic DNA samples. We prepared four

independent replicate libraries with NA12878 genomic DNA and one individual library each for the Ashkenazi Jewish trio. We used the KAPA HyperPlus Kit to prepare the libraries, following the manufacturer's guidelines and then quantified on Qubit (Invitrogen) and verified on Agilent 2100 Bioanalyzer (Agilent Technologies). The libraries were then sequenced on a HiSeq X Ten (Illumina) producing paired reads of 150 nt at the Kinghorn Centre for Clinical Genomics, Darlinghurst, NSW.

To observe the impact of alternative methods and technologies on the DNA ladder, we also performed nanopore sequencing, target-enriched sequencing, and a library of fragmented DNA. For nanopore, the libraries were prepared with a LSK108 kit (1D ligation) according to the manufacturer's instructions and one of them was analyzed on a MinION instrument while the other on a PromethION instrument, at the Kinghorn Centre for Clinical Genomics, Darlinghurst, NSW. Base calling was achieved using ONT Albacore Sequencing Pipeline Software (version 1.2.6).

For the capture sequencing, we used a custom gene panel manufactured by Roche-NimbleGen that contains probes targeting sequence elements of the DNA ladders. Four libraries were prepared with a KAPA Library Preparation Kit (Illumina platform KR0935—v2.14), in conjunction with SeqCap Adapter Kits (Roche-NimbleGen), as per the manufacturer's protocol, with ten cycles of PCR amplification. Purified libraries were quantified on an Agilent 2100 Bioanalyzer, before performing paired-end sequencing on a HiSeq 2500, as above.

Finally, we built a library with median DNA fragment size of 165 nt using the KAPA kit. The purified library was quantified using Qubit (Invitrogen) and the size verified on Agilent 2100 Bioanalyzer with the Agilent High Sensitivity DNA Kit (Agilent Technologies). We then sequenced the library a HiSeq 2500, as above.

**DNA ladder bioinformatic analysis**. To minimize any imprecision of mixing DNA ladders in the laboratory, we partitioned reads originating from each individual DNA ladder in a sequencing library according to their sequence, and then subsampled each of them to the same depth of coverage. This omitted potential errors carried out by inaccurate quantification of DNA or pipetting biases.

We used Jellyfish "count" (version 2.2.10) to quantify k-mers in simulated and experimental libraries containing the DNA ladder[29]. The k-mer length was 31 and we also applied the canonical option (-C) to count equivalent sequences in forward or reverse orientation as the same. We then extracted the counts for DNA ladder k-mers or any other k-mers of interest with Jellyfish "query" (version 2.2.10). For DNA ladder k-mers, we matched the observed counts with the expected cn and quantified variation at different cn units.

**Modeling the impact of different technical variables on the DNA ladder**. We evaluated the impact of different technical variables on quantification of the DNA ladder. We used the DNA ladder linear regression slope (k-mer count versus expected cn) as an indicator of quantitative resolution. We also calculated the coefficient of variation as a standardized measurement of variability between the different cns in the DNA ladder within and between sequencing libraries. We modeled sequencing error, library depth, and library complexity (through read duplication).

First, we simulated sequencing libraries using the Wgsim software (version 1.9), with DNA ladders sequences as templates, but varying the sequencing error rate between 0.0 and 0.1 by intervals of 0.005. We then quantified DNA ladder k-mers and calculated the regression slope and coefficient of variation for each cn unit in the different DNA ladders. We then plotted the relationship between error rate and slope and also error rate and coefficient of variation for the different cn in the DNA ladder.

Next, we subsampled an experimental library, consisting of a neat mixture of the DNA ladder, to varying fractions of the original library depth. We used the "seqtk sample" tool (version 1.0-r82-dirty[30]) and subsampled the library by 5% to a lower limit of 10% of the original library size. We then similarly quantified DNA ladder k-mers and calculated the regression slope and coefficient of variation for each cn unit in the different DNA ladders. As a result, we plotted the relationship between sequencing depth (represented as subsampling fraction) versus slope and also coefficient of variation for the different cn in the DNA ladder.

We emulated read duplication in the composition of a simulated sequencing library (with no sequencing error). Initially a library of unique sequencing reads was generated. Then, for a given duplication rate ($x$), 1-$x$ reads were subsampled from the pool of unique reads, while $x$ reads were chosen (with replacement) from that set of subsampled reads, making sure they were duplicated in the final library. We simulated from 10 to 95% of duplication with 5% incremental increases. The resulting libraries were quantified and the duplication rate was plotted relative to slope and also coefficient of variation for the different cn in the DNA ladder.

**Using the DNA ladder to detect quantitative fold differences**. For samples that are at comparable sequencing depth or normalized, due to the constant nature of the DNA ladder, we can estimate the degree of technical variation between them by calculating the observed difference in counts for DNA ladder k-mers. The distributions for successive cn units were centered around 0 and the standard deviation was used as an estimate of technical variability between the samples. They indicate the amount of variation in counts that should be expected between the samples at different cns. Therefore, for a given significance level (i.e., 0.05) it is possible to derive cut-off limits, or perform a one-sided $t$-test, for observed difference in counts that exceed what is expected for technical variability at a given cn. Similarly based on the distributions, one can calculate the probability that a given difference in count is observed between samples.

**Experimental and simulated mock microbial communities**. Genomic DNA was extracted from nine different bacterial species, quantified using Nanodrop (Thermo Scientific) and the integrity evaluated in an agarose gel. Genomic DNA was then combined into three different mixtures (Supplementary Table 1). The DNA ladder was spiked into the microbial mixtures at 1% fractional abundance. Libraries were prepared with the Nextera XT Sample Prep Kit (Illumina®) according to the manufacturer's instructions. Prepared libraries were quantified on a Qubit Fluorometer (Life Technologies) and verified on an Agilent 2100 Bioanalyzer with an Agilent High Sensitivity DNA Kit (Agilent Technologies). They were then

sequenced on a HiSeq 2500 instrument (Illumina®) producing paired reads of 125 nt at the Kinghorn Centre for Clinical Genomics, Sydney, Australia.

The resulting libraries were mapped to a combined reference containing the assembled genomes of bacteria present in the mixture (RefSeq IDs provided in Supplementary Table 1). We then partitioned the genomes in 100,000 nt windows and combined mapped reads at different abundances between the three mixtures (Supplementary Table 2). This resulted in three different mock communities of bacterial DNA (A, B, and C) with known fold differences between them. We quantified the libraries by performing k-mer counting with Jellyfish "count" (version 2.2.10). The sum of log fold changes for k-mers in the communities A and B were unbalanced or different from 0, while the log fold changes for k-mers in communities A and C were balanced or equal to 0.

Besides generating mock communities (A, B, and C) using experimental reads, we also made simulated libraries for each of them. We used the same 100,000 nt windows as templates and maintained the exact same relative proportions between the communities. They also had the same sizes of the libraries generated with experimental reads. The simulation was performed using the Wgsim software (version 1.9).

**Data normalization**. To assess the impact of data normalization on the detection of fold differences we compared communities A–B and A–C at the k-mer level. With both simulated and experimental reads the library for community A was subsampled to 60% of the original size to create an imbalance with the other communities and make normalization relevant. For each pairwise comparison we applied the following normalization methods directly on the samples k-mer counts: MR (DeSeq2—version 1.24.0), TMM (edgeR—version 3.26.0), and UQ (custom function). Alternatively, these normalizations were first applied on the DNA ladder k-mers and by comparing unnormalized and normalized counts, a scaling factor was derived for each sample, which was then applied to all bacterial k-mers. The detection of fold differences was performed with a one-sided $t$-test using the standard deviation estimated from the synthetic ladder.

**Statistical analysis and graph plotting**. R (v3) was used to generate plots and perform statistical calculations presented in figures and the main text.

**Reporting summary**. Further information on research design is available in the Nature Research Reporting Summary linked to this article.

## Data availability
Associated data files for whole-genome sequencing and metagenomics analyses can be downloaded from SRA (BioProject Accession numbers: PRJNA625156 and PRJNA625162) and any other synthetic sequences, variant annotations and sequencing libraries can be accessed at http://www.sequinstandards.com and https://github.com/almreis/Synthetic_Ladder. Source data are provided with this paper.

## Code availability
Custom code used in the analysis is available at https://github.com/almreis/Synthetic_Ladder and analytical software to automatically process DNA ladders can be found at www.sequinstandards.com.

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

## Acknowledgements
We acknowledge the following funding sources: National Health and Medical Research Council (NHMRC grant nos. APP1108254, APP1114016, APP1136067), UNSW Tuition Fee Scholarship (TFS; to A.L.M.R) and Cancer Institute NSW Early Career Fellowship 2018/ECF013 (to I.W.D.). The contents of the published materials are solely the responsibility of the administering institution, a participating institution or individual authors, and they do not reflect the views of the NHMRC or CINSW.

## Author contributions
A.L.M.R, I.W.D, and T.R.M. conceived the project and devised the experiments. B.S.M, C.B., and J.B. prepared and sequenced NGS libraries, and conducted laboratory experiments. E.M. provided and prepared mock microbial communities. A.L.M.R. and T.W. performed data analysis. A.L.M.R, I.W.D., and T.R.M. prepared the paper, with support from all co-authors.

## Competing interests
The Garvan Institute of Medical Research has filed patent on techniques described in this study (US Patent App. 15/535,768). The authors declare no competing interests.
