## [Peer Review File · Nature Communications]

Reviewers' Comments:

Reviewer #1:

Remarks to the Author:

SUMMARY

Reis et al describe and test a spike-in control DNA molecule that can be used to help scale / normalize quantitative measurements in varied genomics applications. The authors call this spike-in control a "synthetic DNA ladder," but the main goal is not to measure DNA length in the traditional sense of a molecular biology ladder, but to scale copy number measurements made via read-mapping and read-depth-based quantification across samples. The major conceptual advance over their prior work is to encode quantitative differences in reference sequences contained within a single DNA molecule (as ~ 600 bp repeats) which should in theory reduce batch-to-batch variation as compared to similar spike-in controls comprised of independent DNA molecules.

Making **accurate** quantitative measurements with high-throughput sequencing is an important topic, in my opinion often overlooked by the ease of naively quantifying by read mapping alone. This study is notable for the many careful experiments the authors used to show the utility of their ladder as compared to quantitative estimates of copy number made without such a spike-in control. I agree with the authors that a potential strength of the approach is the simplicity of having a single molecule (or 15) for the spike in. However, I have some perhaps surprising concerns about the way the study is justified and presented. Most notably, there is essentially no justification for the study provided in the introduction, and very little evidence provided that this particular spike-in control is superior to others present in the literature. I worry that without addressing these concerns, it is hard to understand the significance of the advance, here. I detail these concerns and others below.

The manuscript is well-written and the figures are mostly intuitive and helpful.

MAJOR COMMENTS

There is essentially no evidence provided in the very brief introduction to justify developing such a spike-in control. In particular, I was surprised to find essentially only one relevant reference. Given this one relevant reference does come from a Nature Genetics review from the corresponding author's lab, a more thorough coverage of the relevant literature of spike-in controls / standards in genomics research is really inexcusable. The 2 paragraphs that set up the study are about 1) the general importance of standards in science and 2) gel electrophoreses ladders. It is completely unclear in the introduction what the state of the field is with regards to spike-in standards for aiding quantitative measurements based on copy-number estimation in genomics. This is by now a maturing field in shotgun metagenomics, at least, (including important contributions from the Mercer lab) and the reader needs to understand why standards are needed in general, and specifically what the deficiencies are in existing approaches that you are hoping to address here. Especially given the varied applications spanning multiple disciplines that the authors address with their experiments, it is important to justify using spike-in standards and explain why the current approaches are insufficient.

Given that the authors (and others) have in the past developed similar spike-in standards, I find it unusual that the comparison to other approaches, including their own, is limited to 3 sentences in the discussion (lines 298-305). Did the authors compare their previous "sequins" (Hardwick et al or Devesin et al) to the current ladder? If not, why, and can they expand on when one version of spike-in control or other should be used. Is the only advance here the one discussed (reducing batch-to-batch variability in preparation)? The refs provided to support this claim (the authors' previous work in this area) don't really show the problem of batch to batch variability. If this is the main advance here over these previous efforts, then this makes it seem like the work here is more incremental, without more justification from the literature that would help understand the

exceptional importance of this. For example, I'm thinking of papers showing lab-to-lab variability or technical variability from the Human Microbiome Project.

In the methods section, there is not enough description of how the standard and particularly the subsequences were chosen / generated. I see figure S1, but what was the algorithm for sequence *design*? Also, why were 15 independent ladders designed? I think this belongs in the methods section.

Figure 2c. This is a fascinating and powerful figure, but I am unclear on a key detail. What exact distributions are plotted across the x axis for each individual treatment (library-prep by sequencing technology)? If it were the mean and sd count for each ladder library molecule, I would anticipate 15 data points per treatment (one per library molecule), but there appear to be many more than 15 per treatment. An expanded description in the legend would help me. Also, consider identifying the treatment shown in Figure 2.a.b. in this panel.

There is no mention in the manuscript of data availability.

line 459: The composition of the bacterial mock communities are not described sufficiently to reproduce this work.

MINOR COMMENTS

Personally, I find the term "ladder" confusing. I have such a strong association for a "ladder" as both measuring length and being composed of independent molecules that I find it unintuitive here. Only if other reviewers agree, you might consider giving your approach another, more accurate name.

Figure 1C: I think it would be more intuitive to maintain the same y scale across panels, to show the difference in slope visually. More importantly, it is unclear what was sequenced in panel c.c. Experimental Library. From the Results, it appears that there should be 15 individual ladders on this figure (the "neat mixture"). Is that true? It might be worth pointing that out in the legend, so that the reader has context for the multiple vectors shown.

Figure 2a.a (left): It appears the red distribution extends into negative k-mer count. Is the x axis correct? Also, in the right panel (2a.b ?) there appear to be some kmers in each of the cn distributions which are greatly underrepresented (e.g. count=0 for 1cn, other bumps in distribution to the left for other cn). There is not a comment to address here for the manuscript, but are these areas of the artificial sequence that can be optimized?

Figure 3a/3b: It is striking (to someone that doesn't work in human genomics) how much overlap there is in the kmer count distributions for diploid and haploid kmers. Nothing in the variant detection experiments of Figure 4 used the sequence composition of the kmers – this is just a t-test informed by the ladder cn distributions. Is there a chance to learn how kmer sequence / sequence composition is related to the observed k-mer count distributions per cn, and use that (for example) to inform variant detection? Do existing variant detection algorithms use sequence composition of the surrounding region? If so, at minimum it might be worth pointing out that "composition corrections" could potentially be learned on a per-run basis from the many kmers in the ladder molecules.

line 16: semicolon after nature.

lines 57-58: "...the read count for each subsequence..." This language is confusing to me. Really, you are looking at kmer coverage, and read lengths differ, reads will span sub-sequence boundaries, etc. I think it would be better to speak here (and elsewhere?) about "coverage."

line 75: "a experimental"  "an experimental"

line 96: I believe you are referring to Figure 1b here, not 1c as written.

lines 100-101: "Due to the additional technical variation introduced during experimental steps" It would help greatly here to describe in one sentence what "technical variation" means / could mean.

line 222: "the dashed line (left) indicates the significance threshold" I believe this should be "(right)" instead of left.

line 268: overestimates  underestimates

lines 339-340 "the final ladder sequence was confirmed on an agarose gel." Does this mean the size of the final ladder sequence was confirmed? Or that you did sanger sequencing. I doubt that the "sequence was confirmed."

Reviewer #2:

Remarks to the Author:

The authors developed a synthetic DNA ladder that can be spiked-in to a DNA sample, serving as an internal scale that measures quantitative genetics features. The ladder is encoded within a single molecule making it easy for synthesis by different labs. Applications to human genome and metagenome are illustrated. While the paper is of interest, it needs to address a few major points below.

1 It would be better to compare the stacked DNA ladder from this work with other previous spike-in controls. For example, check if this ladder dramatically outperformed the simple spike-in with a single constant copy number, e.g. lambda DNA?

2 The authors described that the sequences of all the DNA ladder sub-elements are non-homology with any known natural nucleotide sequences. However, how about the sequence identities across the sub-elements from the 15 synthetic ladders at inter- and intro-ladder level? If there's no significant identity overlaps, how to explain the overlap between $cn_{1/2}$ in experimental library (Fig. 2a)?

3 In the reference-free human genome analysis part, the author demonstrated the ladder can accurately measure quantitative genetic features. Regarding the low range (8 times) of this ladder, it will be better to check out some rare variant alleles out of the aforementioned range. While the metagenomic application makes great sense (normalizing between libraries), it is unclear it is necessary to use this ladder for human genome research.

4 For Nanopore data, was the same input ladder amount used as other sequencing platforms? The mapped reads from nanopore data are significantly less than other groups. Additionally, was the same k-mer length (31 mer) used for nanopore data? Considering the features of Nanopore data (longer read length and low accuracy), maybe increase the k-mer size can get better results?

REVIEWER 1

Major comments:

1.1 There is essentially no evidence provided in the very brief introduction to justify developing such a spike-in control. In particular, I was surprised to find essentially only one relevant reference. Given this one relevant reference does come from a Nature Genetics review from the corresponding author's lab, a more thorough coverage of the relevant literature of spike-in controls / standards in genomics research is really inexcusable. The 2 paragraphs that set up the study are about 1) the general importance of standards in science and 2) gel electrophoreses ladders. It is completely unclear in the introduction what the state of the field is with regards to spike-in standards for aiding quantitative measurements based on copy-number estimation in genomics. This is by now a maturing field in shotgun metagenomics, at least, (including important contributions from the Mercer lab) and the reader needs to understand why standards are needed in general, and specifically what the deficiencies are in existing approaches that you are hoping to address here. Especially given the varied applications spanning multiple disciplines that the authors address with their experiments, it is important to justify using spike-in standards and explain why the current approaches are insufficient.

Dear Reviewer, please find below our revised introduction, which provides additional background on the use of reference standards in genomics as requested:

“Scientific enquiry relies on standard units of measurement to provide precise, reproducible and quantitative descriptions of natural phenomena [1]. These measurements are often taken relative to reference standards with known, reliable and often graduated properties [2, 3].

A commonly used reference standard in molecular biology is the DNA ladder, which constitutes of a set of DNA fragments of known sizes that forms a ladder during migration in gel electrophoresis [4]. This DNA ladder provides a reference to determine the size of other unknown DNA fragments migrating in the gel. The use of a common ladder also allows the size of DNA fragments to be compared between experiments, and under different migration conditions.

Despite these advantages, no similar quantitative DNA ladder is available against which to measure abundance in next-generation sequencing. Natural genetic materials, such as reference human genomes or small phage genomes, do not provide a graduated scale [5-7]. Synthetic spike-in controls can be mixed at known concentrations to form a quantitative ladder [8, 9]. However, errors during mixing and preparation can result in an inconsistent and inexact graduated scale that is unable to directly measure technical variation. Furthermore, these irregular errors result in differences between batches of spike-in controls.

To address these limitations, we have designed a quantitative DNA ladder that is encoded within a single synthetic DNA sequence. This single ladder sequence encodes multiple contiguous sub-sequences, each repeated at known copy-numbers. When sequenced, the read count from each sub-sequence is proportional to its copy-number within the ladder, and in a known and exact ratio to other sub-sequences within the ladder. Together, these sub-sequences form an exact scale across a dynamic quantitative range within an NGS library.

Given that no mixing is required to assemble the ladder, any observed deviation from the expected quantitative scale indicate the scale of technical variation resulting from experimental sources, such as library preparation bias, sampling and coverage depth and sequencing error. Therefore, in addition to measure quantitative abundance, the ladder can also measure technical uncertainty or variability within the NGS library

The synthetic DNA ladder is comprised of artificial sequences and is independent from any reference, and can be universally applied in different samples or applications. Within this study, we use ladders in metagenome analysis and human genome sequencing, where they provide a quantitative framework for reference-free genome analysis. Finally, we show how the synthetic ladder achieves best-in-class library normalisation, and thereby enables more accurate quantitative comparisons between samples and more efficient data sharing between laboratories.”

1.2 Given that the authors (and others) have in the past developed similar spike-in standards, I find it unusual that the comparison to other approaches, including their own, is limited to 3 sentences in the discussion (lines 298-305). Did the authors compare their previous “sequins” (Hardwick et al or Deveson et al) to the current ladder? If not, why, and can they expand on when one version of spike-in control or other should be used. Is the only advance here the one discussed (reducing batch-to-batch variability in preparation)? The refs provided to support this claim (the authors' previous work in this area) don't really show the problem of batch to batch variability. If this is the main advance here over these previous efforts, then this makes it seem like the work

here is more incremental, without more justification from the literature that would help understand the exceptional importance of this. For example, I'm thinking of papers showing lab-to-lab variability or technical variability from the Human Microbiome Project.

We summarise the major advantages of the ladder over previous spike-ins (such as sequins) as follows:

1. The DNA ladder provides an exact, known and constant quantitative scale. Other spike-ins (such as sequins) form a quantitative scale by mixing individual DNA molecules at different concentrations. This approach cannot achieve the same level of accuracy due to errors in mixture and preparation (DNA quantification, dilution, pipetting, etc). To illustrate the difference in accuracy between the DNA ladder and other spike-ins, we show the observed relative to expected abundances of the DNA ladder compared to spike-in controls (**Response Figure 1a,b**). This shows the DNA ladder more accurately represents the original stoichiometries than the spike-ins.

2. The DNA ladder enables technical variation to be determined across quantitative range. Technical variation observed in spike-in ladders is due to experimental variation and mixing errors. Because no mixing is required to assemble the DNA ladder, technical variation observed in the DNA ladder is only due to experimental variation (such as sequencing errors, sampling bias etc.).

To demonstrate this distinction, we measured the proportional variation in observed abundances of the DNA ladder and the spike-ins (**Response Figure 1c**). We see that the technical variation of the spike-ins is much greater due to confounding additional mixing errors. As a result, we cannot accurately measure the additional technical variation due to NGS library preparation and sequencing. In contrast, the technical variation measured from the ladder is not impacted by mixing errors.

This ability to estimate technical variation from the ladder is a key advantage that enables a wide range of new analytical possibilities. For example, we can perform a t-test between samples to estimate significant differences between two single genomes (as demonstrated in the Section "Reference-free human genome analysis with the DNA ladder") which is not otherwise possible with spike-in controls.

3. The final advantage (as indicated by the reviewer) is that the DNA ladder does not suffer from batch to batch variation. Because the ladder is encoded within a single sequence, the design can be easily shared and independently prepared by different laboratories. To demonstrate this, we have shown the performance (quantitative accuracy, technical variation etc.) from two independent DNA ladder and spike-in NGS libraries (**Response Figure 1a-e**). This shows that batch-specific differences apparent between the spike-in preparations do not similarly afflict the DNA ladder.

The analysis and figure described above have now been included in Supplementary Materials and the text (page 5) has been amended as follows:

"We finally compared the performance of the synthetic DNA ladder to spike-in controls previously designed for use in metagenomics (**Figs. S2a and S2b; [9]**). Unlike the DNA ladder, these spike-in controls form a quantitative scale by mixing individual DNA molecules at different concentrations, which can result in mixing errors. We see that the proportional variability at different copy-numbers in the ladder is on average 2-fold lower compared to the spike-ins (**Figs. S2c and S2d**). Furthermore, the DNA ladder also has more accurate ratios between subsequent copy-numbers within the ladder compare to the spike-ins (**Figs. S2e**). This demonstrates DNA ladder provides a more exact quantitative scale than comparable spike-in controls, and eliminates confounding technical variables that are often introduced during mixture preparation of other comparable spike-in controls."

Figure 1. (a) Scatter plots show the observed abundance (in median k-mer counts) of sequence elements versus expected copy-number in the synthetic DNA ladder. Two independent libraries are shown (Sample1=orange and Sample2=green); (b) Scatter plots show the observed abundance (in median k-mer counts) of individual spike-ins [9] versus expected copy-number; (c) Boxplot graphs show the k-mer count distribution for two libraries (containing the synthetic DNA ladders, spike-in controls) that were independently prepared and sequenced to differing depths; (d) Scatter plot shows the coefficient of variation at different copy-numbers for the ladder and spike-ins when considering all k-mer counts in the two independent libraries, with the dashed lines indicating the corresponding means. (e) The plot shows the median ratios between subsequent copy-numbers for individual synthetic DNA ladders and spike-ins in the two independent libraries, with horizontal line indicating the mean.

1.3 In the methods section, there is not enough description of how the standard and particularly the subsequences were chosen / generated. I see figure S1, but what was the algorithm for sequence *design*? Also, why were 15 independent ladders designed? I think this belongs in the methods section.

We have expanded the description of how the ladder sequences were generated below. Briefly, we designed 15 independent ladders to enable a larger k-mer representation, and to mitigate the impact of biases or errors that might affect individual ladders. It was also anticipated that some ladders would fail during synthesis, validation etc. We have incorporated a more detailed description of the ladder design in the Methods section (page 15) as follows:

“Each ladder has 4 different sub-sequence elements at different copy-numbers. We set the length of sub-sequence elements to 600nt so they were longer than the typical read or fragment length observed in short-read library preparation protocols. To obtain the sub-sequences, we selected 600nt long sequences from hg38 which were the shuffled to remove homology. To confirm the artificiality of our sequences, we performed a *blast* search against the non-redundant nucleotide database (nr/nt [10], which found no exact matches longer than 25nt. We randomly organised 4 sub-sequences into 15 ladders by joining the sub-sequences according to the copy-numbers 1, 2, 4 and 8. An intervening unique artificial sequence of 20nt was added between each sub-

sequence element and a 500nt common sequence was added to 5' and 3' end of each ladder to mitigate potential edge effects.”

1.4 Figure 2c. This is a fascinating and powerful figure, but I am unclear on a key detail. What exact distributions are plotted across the x axis for each individual treatment (library-prep by sequencing technology)? If it were the mean and sd count for each ladder library molecule, I would anticipate 15 data points per treatment (one per library molecule), but there appear to be many more than 15 per treatment. An expanded description in the legend would help me. Also, consider identifying the treatment shown in Figure 2.a.b. in this panel.

This illustration aimed to provide a simple and intuitive representation of the DNA ladder performance under different experimental conditions, and is analogous to the DNA ladders used in gel electrophoresis (which were our original source of insight).

We revised the description of **Figure 2** (page 6) to clarify the distributions are plotted across the x axis as follows:

“Figure 2. (c) The panel illustrates the variability associated with cn units across a range of libraries generated with different sequencing technologies (e.g. ONT Nanopore MinION, ONT Nanopore PromethION, Illumina HiSeq X Ten, Illumina NextSeq 500 and Illumina HiSeq 2500 instruments) or prepared with alternative protocols (e.g. KAPA, KAPA HyperPlus (PCR-free and PCR-based) and Nextera XT kits, or target enrichment by oligonucleotide hybridisation). *The y-axis indicates k-mer counts, whilst the x-axis indicates the position across the sub-sequence length (600nt). The mean k-mer count (opaque line) and standard deviation (transparent line) are determined from all 15 ladders.*”

1.5 There is no mention in the manuscript of data availability.

All NGS libraries are being uploaded to the Sequence Read Archive (SRA; <https://www.ncbi.nlm.nih.gov/sra>). Provisional accession identifiers are SUB6850669 and SUB7249805. Data will also be hosted independently at www.sequinstandards.com/resource.

1.6 line 459: The composition of the bacterial mock communities are not described sufficiently to reproduce this work.

The composition of the mock microbial communities has now been provided as an Appendix and the main text (page 18) has been amended as follows:

“Genomic DNA was extracted from 9 different bacterial species, quantified using Nanodrop (Thermo Scientific) and the integrity evaluated in an agarose gel. Genome DNA from the different species was then combined into three different mixtures (**Supplementary Table1**). The DNA ladder was spiked into each microbial mixtures at 1% fractional abundance. Libraries were prepared with the Nextera XT Sample Prep Kit (Illumina®) according to the manufacturer’s instructions. Prepared libraries were quantified on a Qubit Fluorometer (Life Technologies) and verified on an Agilent 2100 Bioanalyzer with an Agilent High Sensitivity DNA Kit (Agilent Technologies). They were then sequenced on a HiSeq 2500 instrument (Illumina®) producing paired reads of 125nt at the Kinghorn Centre for Clinical Genomics, Sydney, Australia.

The resulting libraries were mapped to a combined reference containing the assembled genomes of bacteria present in the mixture (RefSeq IDs provided in **Supplementary Table2**). We then partitioned the genomes in 100,000nt window and combined mapped reads at different abundances between the three mixtures (**Supplementary Table2**). This resulted in three different mock communities of bacterial DNA (A, B and C) with known fold-differences between them. We quantified the libraries by performing k-mer counting as described above. The sum of log fold-changes for k-mers in the communities A and B were unbalanced or different than 0, while the log fold-changes for k-mers in communities A and C were balanced or equal to 0.

Besides generating mock communities (A, B and C) using experimental reads, we also made simulated libraries for each of them. We used the same 100,000nt windows as templates and maintained the exact same relative proportions between the communities. They also had the same sizes of the libraries generated with experimental reads. The simulation was performed as previously described.”

Minor comments:

1.7 Personally, I find the term “ladder” confusing. I have such a strong association for a “ladder” as both measuring length and being composed of independent molecules that I find it unintuitive here. Only if other reviewers agree, you might consider giving your approach another, more accurate name.

We have tried to use alternative terms, such as ‘scale’ and ‘ruler’, but they weren’t well received. We can alternatively add a qualifier, such as *single molecule ladder* for consideration.

1.8 Figure 1C: I think it would be more intuitive to maintain the same y scale across panels, to show the difference in slope visually. More importantly, it is unclear what was sequenced in panel c.c. Experimental Library. From the Results, it appears that there should be 15 individual ladders on this figure (the “neat mixture”). Is that true? It might be worth pointing that out in the legend, so that the reader has context for the multiple vectors shown.

We have re-plotted the graphs in **Figure 1c** to have the same axis (1-400). In addition, we have revised the legend (page 4) to indicate that multiple ladders are within the neat mixture as follows:

“**Figure 1. (c; top panel)** Scatter-plots indicate the observed abundance (in k-mer counts) of sequence elements of 15 different synthetic DNA ladders versus copy-number in three different contexts: a hypothetical ‘perfect’ library, a simulated library and an experimental library. **(bottom panel)** Histograms show the ratios between subsequent copy-numbers in the 15 synthetic DNA ladders that were manufactured.”

1.9 Figure 2a.a (left): It appears the red distribution extends into negative k-mer count. Is the x axis correct? Also, in the right panel (2a.b ?) there appear to be some kmers in each of the cn distributions which are greatly underrepresented (e.g. count=0 for 1cn, other bumps in distribution to the left for other cn). There is not a comment to address here for the manuscript, but are these areas of the artificial sequence that can be optimized?

We thank the reviewer for identifying this error in **Figure 2a**. The k-mer distribution should not be negative and has been corrected. K-mers with 0 counts in the experimental library indicate a minority of events wherein a DNA mutation within the synthetic ladder has abolished the k-mer. These have also been removed from **Figure 2a**.

1.10 Figure 3a/3b: It is striking (to someone that doesn’t work in human genomics) how much overlap there is in the kmer count distributions for diploid and haploid kmers. Nothing in the variant detection experiments of Figure 4 used the sequence composition of the kmers – this is just a t-test informed by the ladder cn distributions. Is there a chance to learn how kmer sequence / sequence composition is related to the observed k-mer count distributions per cn, and use that (for example) to inform variant detection? Do existing variant detection algorithms use sequence composition of the surrounding region? If so, at minimum it might be worth pointing out that “composition corrections” could potentially be learned on a per-run basis from the many kmers in the ladder molecules.

We agree that the overlap between the 1cn (haploid) and 2cn (diploid) k-mers is substantial and it is presumed this is the result of two variables (i) sequence composition (as highlighted by the reviewer) and (ii) experimental variation (sequencing errors etc.). Notably, we show that by increasing experimental variation (by reducing library depth) we observe an increase in the overlap between 1cn and 2cn. We have not used the k-mer profile from the synthetic DNA ladders to perform quantitative normalisation based on their sequence. However, we are currently developing improved base-calling algorithms that are learned on a per-run basis using synthetic controls. We are finding this approach is particularly suited for improving the base-calling accuracy of nanopore sequencing (Oxford Nanopore Technologies), and aim to publish these new results soon.

As suggested by the reviewer, we have highlighted this potential use of DNA ladders in the discussion (page 13) as follows:

“Additionally, given the diverse range of k-mers present in the synthetic ladders, they may be used as an in-run reference against to measure sequencing errors, improve base-calling algorithms and normalise sequencing biases.”

1.11 line 16: semicolon after nature.

This change was incorporated in the text.

1.12 lines 57-58: “...the read count for each subsequence...” This language is confusing to me. Really, you are looking at kmer coverage, and read lengths differ, reads will span sub-sequence boundaries, etc. I think it would be better to speak here (and elsewhere?) about “coverage.”

The change was incorporated in the text.

1.13 line 75: "a experimental"  "an experimental"

The change was incorporated in the text.

1.14 line 96: I believe you are referring to Figure 1b here, not 1c as written.

The change was incorporated in the text.

1.15 lines 100-101: "Due to the additional technical variation introduced during experimental steps" It would help greatly here to describe in one sentence what "technical variation" means / could mean.

The text has been amended (page 3) as follows:

"We next demonstrated the performance of the synthetic DNA ladder in an experimental NGS library that is subject to technical variation. Sources of variation include sample amplification, library preparation biases and sequencing errors that accumulate in an experimental protocol. We prepared and sequenced an NGS library from the neat mixture of synthetic DNA ladders, and then plotted the observed k-mer counts for each sequence element relative to the expected cn. Due to the additional technical variation introduced during experimental steps, the observed DNA ladder had a weaker linear relationship ($R^2=0.9605$) and greater variation in median cn ratios (mean=2.034, sd=0.3950) than the simulated library (Fig. 1c)."

1.16 line 222: "the dashed line (left) indicates the significance threshold" I believe this should be "(right)" instead of left.

The change was incorporated in the text.

1.17 line 268: overestimates  underestimates

The change was incorporated in the text.

1.18 lines 339-340 "the final ladder sequence was confirmed on an agarose gel." Does this mean the size of the final ladder sequence was confirmed? Or that you did sanger sequencing. I doubt that the "sequence was confirmed."

The text has been amended (page 15) as follows:

"The plasmids containing the ladders were transformed in *E. coli*, then grown in a 50 ml culture and later purified. Each ladder excised from the plasmids; then the size of the final ladder sequence was confirmed on an agarose gel and quantified by UV fluorometry (ThermoFisher Qubit). In total 15 different DNA ladders were generated."

REVIEWER 2

Major comments:

2.1 It would be better to compare the stacked DNA ladder from this work with other previous spike-in controls. For example, check if this ladder dramatically outperformed the simple spike-in with a single constant copy number, e.g. lambda DNA?

Single molecule spike-in controls, like *lambda phage* and *phiX*, can provide a simple spike-in control to measure sequencing errors etc. However, they do not provide a ladder across a dynamic quantitative range. Therefore simple spike-ins cannot be used as a reference materials for measuring quantitative genome features.

However, we can compare the DNA ladder to other spike-in control mixtures, wherein the individual spike-ins are added at different concentrations across a quantitative range. For example, we have compared the DNA ladder to metagenome spike-ins [9], where we show that the DNA ladder has greater quantitative accuracy and less variation than those spike-in controls (see **Response 1.2 for further details**).

2.2 The authors described that the sequences of all the DNA ladder sub-elements are non-homology with any known natural nucleotide sequences. However, how about the sequence identities across the sub-elements from the 15 synthetic ladders at inter- and intro-ladder level? If there's no significant identity overlaps, how to explain the overlap between cn1/2 in experimental library (Fig. 2a)?

There is no significant homology within or between ladder sequences. To demonstrate this, we compared the ladder sequences, as well as 10 random human (hg38) sequences, showing all sequences to be equally dissimilar (mean=0.74; sd=0.02; **Response Figures 2a,b**).

The overlap between 1cn and 2cn identified by the reviewer in the experimental library is not due to sequence similarity, but instead due to technical variation from experimental sources (such as depth of coverage or sequencing errors). This overlap indicates the fraction of 1 and 2 copy k-mers that will be of indistinguishable abundance due to the technical variation within the experimental library (**Response Figures 2c,d**).

To illustrate this, we increased the technical variation within the library by reducing the library depth (and thereby increasing variation due to low sampling frequency). When technical variation increases, we observe that the overlap increases (**Response Figures 2e,f**).

This analysis and figure have now been included in **Supplementary Materials** and the text (page 5) was amended as follows:

“This difference between observed and expected k-mer abundances within synthetic DNA ladder indicates technical variation or uncertainty that accumulates during experimental steps of library preparation and sequencing. This is apparent in the symmetric, unimodal distribution of k-mer counts at each level of cn, which is broader in the experimental library than the simulated library (**Fig. 2a**). We also observe less uncertainty at higher levels of cn, reflecting their greater sampling coverage and confidence (CV; 1cn = 0.16, 2cn = 0.12, 4cn = 0.08, 8cn = 0.05; **Fig. 2a**). The intersections between the distributions indicate the fraction of k-mers with counts that do not discriminate between successive copy-numbers (**Fig. S3**). The overlap, which is also indicative of uncertainty, increases with lower depth of coverage or increased sequencing error (**Fig. S3**).”

Figure 2. (a) Matrix comparison of nucleotide sequence between all sub-sequence elements in the synthetic ladders (1, 2, 4 and 8cn) and also ten human sequences of equal length (600nt). Each nucleotide is encoded with a different color; (b) Pairwise matrix comparison showing the distance between sub-sequence elements in the synthetic ladders and hg38 sequences, where distance can range from 0 (identical) to 1 (completely dissimilar). (c and d) Density distribution of k-mer counts for each cn unit (1cn=red, 2cn=yellow, 4cn=green and 8cn=blue) in the ladder. The fraction overlapping between successive copy-number units is indicated in gray. (e and f) The fraction of k-mers overlapping between subsequent copy-numbers decreases relative to increasing variation due to decreasing library depth (by subsampling).

2.3 In the reference-free human genome analysis part, the author demonstrated the ladder can accurately measure quantitative genetic features. Regarding the low range (8 times) of this ladder, it will be better to check

out some rare variant alleles out of the aforementioned range. While the metagenomic application makes great sense (normalizing between libraries), it is unclear it is necessary to use this ladder for human genome research.

We agree that the ladder has limited applications in genome sequencing and the identification of germline variants where current tools are sufficient. Instead, the aim of the “Reference-free human genome analysis” section (page 8) is to validate the quantitative accuracy of the ladder against high-confidence reference materials.

The reference human genome samples harbor a large number of germline variants present at known diploid and haploid abundances. These consanguineous variants are also shared at known copy ratios between these three genomes. Therefore, these genomes comprise a large, high-confidence dataset of k-mers at known and differing quantities, which is ideal to validate the quantitative accuracy and fidelity of 1cn and 2cn within the ladder

We agree that the use of the ladder is more suited to detect larger quantitative variation such as copy-number variants, somatic mutations and complex chromosomal changes in cancer (as highlighted by the reviewer). However, we would also like to highlight that reference-free analysis, where the synthetic ladder is useful, is becoming increasingly popular to address genetic variation in regions where the reference genome is incomplete or when variation is highly divergent to the reference genome.

We have revised the following text (page 9) to indicate this:

“To demonstrate whether the ladder can be used to identify significant biological differences between genomes and in the absence of technical replicates, we next performed whole-genome sequencing of the Ashkenazi Jewish family trio; NA24149 (father), NA24143 (mother) and NA24385 (son) [11]. The resulting libraries were sequenced at comparable depth, and the ladder calibrated to the human diploid genome (**Fig. S6**). These human genomes samples share a large number of germline variants present at known diploid and haploid abundances, which have been extensively characterized with high-confidence. Therefore, these genomes comprise a large, reliable dataset with both control and test k-mers at same and differing quantities, which can be used to validate the quantitative accuracy and fidelity of 1cn and 2cn within the ladder.”

2.4 For Nanopore data, was the same input ladder amount used as other sequencing platforms? The mapped reads from nanopore data are significantly less than other groups. Additionally, was the same k-mer length (31 mer) used for nanopore data? Considering the features of Nanopore data (longer read length and low accuracy), maybe increase the k-mer size can get better results?

The same ladder mixture and input amounts were used for all the different sequencing platforms, including Oxford Nanopore. The same k-mer length (31-mer) was also used for analysing the nanopore data. As noted by the reviewer, this k-mer length results in the Oxford Nanopore libraries appearing to have a lower abundance due to a higher sequencing error rate that abolishes expected ladder 31-mers.

As suggested by the reviewer, we investigated the impact of different k-mer size (10,15,20,25,30,25 and 40nt) on the quantification of the DNA ladder in Nanopore (and Illumina libraries). As expected, as k-mer size decreases, the observed counts for the different ladder units increases (**Response Figure 3a**). In nanopore libraries, we find that decreasing to a 10-mer results in an almost 4-fold increase in observed counts (**Response Figure 3b**). Accordingly, we agree that decreasing k-mer size can provide better results for Nanopore data, albeit at lower specificity and computational speed.

This analysis and figure have now been included in **Supplementary Materials** and the text (page 7) was amended as follows:

“The synthetic ladder can measure the quantitative performance of the different library preparation methods, with a full and detailed evaluation provided in **Fig. S6**. For example, the Illumina Nextera XT™; preparation, which uses lower input DNA amounts and additional amplification steps, exhibits a lower slope and higher variability in cn counts compared to alternative PCR-free library preparation methods that also exhibit stronger linearity (**Figs. S4 and S6**) [12]. Nanopore libraries, where sequencing error is higher, also exhibit a reduced ladder slope (6.49 and 4.22 vs 35.03; **Fig. S6b**) and increased count-variability (**Fig. S6e**) [13]. However, reducing the k-mer length mitigates the impact of sequencing error, and achieves better quantitative performance with Nanopore long-read data (**Fig. S7**).”

a. Impact of k-mer length on ladder quantification in Illumina and Nanopore libraries.

b. Relationship between k-mer length and count per copy-number in Illumina and Nanopore libraries.

Figure 3. (a) Boxplots show the k-mer count distribution for the different cn units in the ladder (1cn=red, 2cn=yellow, 4cn=green and 8cn=blue) in Illumina and Nanopore libraries as the size of k increases. (b) The plot shows the linear relationship between k-mer length and the median observed count for the different cn units in Illumina and Nanopore libraries.

REFERENCES

1. Schadow, G., et al., *Units of measure in clinical information systems*. Journal of the American Medical Informatics Association, 1999. **6**(2): p. 151-162.
2. Radin, N., *What is a Standard?* Clinical chemistry, 1967. **13**: p. 55--76.
3. Hardwick, S.A., I.W. Deveson, and T.R. Mercer, *Reference standards for next-generation sequencing*. Nature Reviews Genetics, 2017. **18**(8): p. 473.
4. Carlson, D.P.a.W., Paul C and Klevan, Leonard, *Size markers for electrophoretic analysis of DNA*. 1994.
5. Manley, L.J., D. Ma, and S.S. Levine, *Monitoring error rates in Illumina sequencing*. Journal of biomolecular techniques: JBT, 2016. **27**(4): p. 125.
6. Zook, J.M., et al., *Integrating human sequence data sets provides a resource of benchmark SNP and indel genotype calls*. Nature biotechnology, 2014. **32**(3): p. 246.
7. Zook, J.M., et al., *An open resource for accurately benchmarking small variant and reference calls*. Nature biotechnology, 2019. **37**(5): p. 561-566.
8. Costea, P.I., et al., *Towards standards for human fecal sample processing in metagenomic studies*. Nature biotechnology, 2017. **35**(11): p. 1069.
9. Hardwick, S.A., et al., *Synthetic microbe communities provide internal reference standards for metagenome sequencing and analysis*. Nature communications, 2018. **9**(1): p. 3096.
10. Altschul, S.F., et al., *Basic local alignment search tool*. Journal of molecular biology, 1990. **215**(3): p. 403-410.
11. Zook, J.M., et al., *Extensive sequencing of seven human genomes to characterize benchmark reference materials*. Scientific data, 2016. **3**: p. 160025.
12. Ring, J.D., et al., *A performance evaluation of Nextera XT and KAPA HyperPlus for rapid Illumina library preparation of long-range mitogenome amplicons*. Forensic Science International: Genetics, 2017. **29**: p. 174-180.
13. Fu, S., A. Wang, and K.F. Au, *A comparative evaluation of hybrid error correction methods for error-prone long reads*. Genome biology, 2019. **20**(1): p. 26.

Reviewers' Comments:

Reviewer #1:

Remarks to the Author:

Thank you for a thorough response to my concerns. I feel all of my comments have been addressed by the authors. I simply note that the final manuscript needs to contain SRA accession numbers for data availability, which the authors have indicated they are in the process of obtaining.

Reviewer #2:

Remarks to the Author:

The authors have addressed my previous comments.